



# First results of the polar regional climate model RACMO2.4

Christiaan T. van Dalum[1], Willem Jan van de Berg[1], Srinidhi N. Gadde[1,2], Maurice van Tiggelen[1], Tijmen van der Drift[1,3], Erik van Meijgaard[3], Lambertus H. van Ulft[3], and Michiel R. van den Broeke[1]

[1]Institute for Marine and Atmospheric Research, Utrecht University, Utrecht, the Netherlands
[2]Faculty of Geo-Information Science and Earth Observation, Twente University, Enschede, the Netherlands
[3]Royal Netherlands Meteorological Institute, De Bilt, the Netherlands

**Correspondence:** Christiaan T. van Dalum (c.t.vandalum@uu.nl)

**Abstract.** A next version of the polar regional climate model RACMO (referred to as RACMO2.4p1) is presented in this study. The principal update includes embedding of the package of physical parameterizations of the Integrated Forecast System (IFS) cycle 47r1. This constitutes changes in the precipitation, convection, turbulence, aerosol and surface schemes, and includes a new cloud scheme with more prognostic variables and a dedicated lake model. Furthermore, the stand-alone IFS radiation
physics module ecRad is incorporated in RACMO, and a multi-layer snow module for non-glaciated regions is introduced. Other updates involve the introduction of a fractional land-ice mask, new and updated climatological data sets, such as aerosol concentrations and leaf-area index, and the revision of several parameterizations specific to glaciated regions. As a proof of concept, we show first results for Greenland, Antarctica and a region encompassing the Arctic. By comparing the results with observations and the output from the previous model version (RACMO2.3p3), we show that the model performs well regarding
the surface mass balance, surface energy balance, temperature, wind speed, cloud content and snow depth. The advection of snow hydrometeors strongly impacts the ice sheet's local surface mass balance, particularly in high-accumulation regions such as southeast Greenland and the Antarctic Peninsula. We critically assess the model output and identify some processes that would benefit from further model development.

## 1 Introduction

Understanding the role of ice sheets in the global climate system is a difficult but essential task. Recent decades show increased mass loss from the Greenland and Antarctic ice sheets (GrIS and AIS, respectively), significantly contributing to contemporary sea level rise (Otosaka et al., 2023; Shepherd et al., 2018, 2020; Dangendorf et al., 2019; Rignot et al., 2019). Increasing summer air temperatures and cloudiness in Greenland have resulted in more surface melt and runoff (Hanna et al., 2021; Noël et al., 2019; Vandecrux et al., 2023), while mass loss by ice-dynamical processes (faster discharge) also remained significant (Khan
et al., 2015; Mottram et al., 2019). Antarctic mass loss is primarily ocean-driven, with enhanced basal melt inducing thinning and/or breakup of ice shelves and accelerating flow of marine-terminating ice streams (Stokes et al., 2022; Gudmundsson et al., 2019; Adusumilli et al., 2020). Several feedback-mechanisms are involved that accelerate changes, such as the snowmelt-albedo feedback (Jakobs et al., 2021; Riihelä et al., 2021).



Contrary to general circulation models (GCMs), regional climate models (RCMs) are developed to investigate specific domains, usually with higher spatial and temporal resolution (Rummukainen, 2010; Belušić et al., 2020). For instance, polar RCMs are used to study the climate of the GrIS and AIS in detail (Van Wessem et al., 2023; Box et al., 2023; Noël et al., 2018; Amory et al., 2021; Van Dalum et al., 2022; Gilbert et al., 2022; Fettweis et al., 2020; Langen et al., 2017) by including additional parameterizations specifically designed to model ice sheets, such as detailed snow albedo schemes and snow metamorphism modules (Van Dalum et al., 2020; Flanner and Zender, 2006). Development of such parameterizations is a continuous process and evaluating their impact on model output is imperative.

The RCM used in this study is the polar (p) version of the Regional Atmospheric Climate Model (RACMO). RACMO simulates the climate and atmosphere-surface interactions of glaciated surfaces and its surrounding areas, and the polar version is used in particular to study the GrIS and AIS (Van Dalum et al., 2021, 2022; Van Wessem et al., 2018, 2021; Noël et al., 2018). Downscaled RACMO output is also applied to smaller glaciated domains like Iceland (Noël et al., 2022) and the model is occasionally run for other domains such as Patagonia (Lenaerts et al., 2014), or used to study processes like the snowmelt-albedo feedback in Antarctica (Jakobs et al., 2021). The model is comprised of two major components: the atmospheric dynamics of the High Resolution Limited Area Model (HIRLAM) and the physics module of the Integrated Forecasting System (IFS) of the European Centre for Medium-Range Weather Forceast (ECMWF). In this study, we introduce a new RACMO version, 2.4p1, that includes, among other changes, an update of the IFS physics from cycle 33r1 (ECMWF, 2009) to 47r1 (ECMWF, 2020). Compared to IFS cycle 33r1, cycle 47r1 includes notable revisions in almost all parts, including the cloud, radiation, convection, turbulence, aerosol, surface and lake schemes. More specifically, cycle 47r1 uses the new aerosol profiles produced by Copernicus Atmospheric Monitoring Service (CAMS) (Bozzo et al., 2020), the Fresh-water Lake (FLake) model (Mironov et al., 2010) and a new radiation scheme ecRad (Hogan and Bozzo, 2016). For some of the parameterizations that are not specifically developed for polar regions, further tuning is required. In addition, several updates are implemented in RACMO2.4p1 that are directly related to the cryosphere and not part of IFS cycle 47r1 or the previous version RACMO2.3p3. This includes a multi-layer snow model for seasonal snow on non-glaciated surfaces (Arduini et al., 2019) and updates to the land-ice mask and blowing snow scheme (Gadde and Van de Berg, 2024).

The aim of this manuscript is to provide a detailed overview of these implemented changes and to evaluate the first results for Greenland and Antarctica. We also highlight the impact that specific parameterizations have on model output and compare results of RACMO2.4 with RACMO2.3p3. This manuscript does not present a new surface mass balance product; this follows in later publications. The manuscript continues with a description of RACMO2.4 and implemented changes (Sect. 2). Methods are described in Sect. 3. Section 4 shows results for Greenland, including cloud content, precipitation, surface mass balance and surface energy balance. Section 5 discusses the surface mass balance and temperature of Antarctica. Additionally, a first impression of RACMO2.4 employed to the Arctic is shown in Sect. 6, where snow depth is discussed. Conclusions and final remarks are made in Sect. 7.



## 2 Description of RACMO2.4

The Regional Atmospheric Climate Model (RACMO) is a hydrostatic model that combines the atmospheric dynamics of HIRLAM (Undén et al., 2002), version 5.0.3 with the physical processes of the ECMWF IFS (ECMWF, 2009). The polar version of RACMO, developed and maintained at the Institute for Marine and Atmospheric Research Utrecht (IMAU), has specialized parameterizations to simulate the climate of polar regions, and has a dedicated glaciated surface tile for which complex snow models and parameterizations are applied. This includes a multi-layer snow module that incorporates processes like snow metamorphism, compaction, melt and refreezing (Ettema et al., 2010; Noël et al., 2018). The last operational version of the polar version of RACMO is 2.3p2 (R23p2) (Noël et al., 2018), followed by a non-operational version 2.3p3 (R23p3) with improved albedo representation (Van Dalum et al., 2020, 2021, 2022).

In this paper, we present a new version RACMO2.4 (R24) that constitutes a major update of the IFS physics module from cycle 33r1 (ECMWF, 2009) to cycle 47r1 (ECMWF, 2020). Furthermore, it also includes all changes already introduced in R23p3. In addition, the coupling between the HIRLAM dynamical core and the IFS physics module is recoded in order to improve code readability and flexibility. Several other aspects of the model that are not related to the IFS physics module are updated as well. Here, we provide an overview of existing code, followed by an overview of changes in R24, starting with updates that are part of IFS cycle 47r1 and then followed by a description of all other updates.

### 2.1 Description of existing code

#### 2.1.1 Dynamical core

R23p3 and R24 use version 5.0.3 of the atmospheric dynamics of HIRLAM (Undén et al., 2002). The HIRLAM dynamical core uses a "two-step" semi-Lagrangian and semi-implicit scheme to determine the origin of an air parcel on a Eulerian grid each time step by integrating the momentum equations on two half steps; first half step integrating the Coriolis term implicitly and the pressure gradient term explicitly, and then the second half step vice versa (McDonald and Haugen, 1992). This allows horizontal advection with unconditionally large time steps. A Raymond filter is applied to smooth orography and lower levels of noise. A number of components from HIRLAM version 6.3.7 are also included, such as the Ritchie-Tanguay (R-T) interpolation to reduce noise on the propagation of the temperature field and smoother precipitation fields near orography (Ritchie and Tanguay, 1996). Since RACMO2.1, precipitation near orography is further reduced by lowering the horizontal diffusion coefficient in the implicit diffusion scheme (Van Meijgaard et al., 2008).

At the lateral boundaries of the model domain, RACMO is forced with multi-level data of wind speed, temperature, humidity and pressure obtained from either a GCM or reanalysis; likewise for sea surface temperature and sea-ice extent at the sea surface boundary. In the simulations performed in the context of this study, the model is forced with ERA5 data (Hersbach et al., 2020). The polar version of RACMO is by default nudged at the upper boundary (Van de Berg and Medley, 2016)





### 2.1.2 Parameterizations adopted from RACMO2.3p3

Several aspects of the model of R23p2 (Noël et al., 2018) have been updated in the evaluated but not yet operational version R23p3 (Van Dalum et al., 2020, 2021, 2022). R23p3 included updates to two major components: a new snow and ice albedo parameterization and changes to the multilayer firn module. The snow and ice albedo schemes are updated by coupling the Two-streAm Radiative TransfEr in Snow Model (TARTES, Libois et al., 2013) to RACMO using the Spectral-to-NarrOWBand ALbedo (SNOWBAL) module version 1.2 (Van Dalum et al., 2019), replacing the broadband albedo scheme of Gardner and Sharp (2010). This allows RACMO to use all 14 shortwave spectral bands available within IFS physics to calculate the albedo. Furthermore, radiative transfer through snow and ice can now occur, resulting in subsurface heating and melt. It also includes firn-module updates, with an increase of vertical resolution and internal melt now creates pore space in ice layers. All aforementioned updates that are part of R23p3 are included in R24.

## 2.2 Updates of RACMO2.4

### 2.2.1 ecRad and aerosols

In R24, the stand-alone version of IFS radiation physics ecRad (version 1.4.1, March 2021) has replaced the previous embedded radiation parameterizations. For the most part, it uses similar routines as before, e.g., the Rapid Radiative Transfer Model (Mlawer et al., 1997), but it also includes bug fixes and noise reduction in heating rates. New parameterizations are also added, such as a new solution to longwave radiative transfer, allowing longwave radiation to scatter in clouds. Liquid cloud optical properties now use the Suite Of Community RAdiative Transfer codes based on Edwards and Slingo (SOCRATES) and ice cloud optical properties follow Baran et al. (2016). The Delta-Eddington approximation is now only applied to particles, as gases do not scatter predominantly in the forward direction. Top of the atmosphere total solar irradiance is changed from 1366 W m$^{-2}$ to a variable number around 1361 W m$^{-2}$ using seasonal variations in Sun-Earth distance and an approximate solar cycle. The 11 climatological monthly-mean CAMS aerosols (Bozzo et al., 2020): organic and black carbon, both hydrophilic and hydrophobic, sulfate and three size bins for sea salt and desert dust, replace the six in the previous scheme of Tegen et al. (1997). Background aerosol optical thickness in the troposphere is changed from 0.03 in IFS cycle 33r1 to 0.05 in cycle 47r1.

### 2.2.2 Clouds and precipitation

A new cloud scheme is introduced with more prognostic variables. Cloud liquid and ice are now separate prognostic variables and can interact with each other. This results in a more sophisticated representation of super-cooled liquid water and supersaturation of liquid and ice clouds. Furthermore, mixed-phase clouds can now occur and supersaturation of ice clouds has become rather common. Rain and snow are now separate and prognostic; they have a fall speed and are transported horizontally by advection. This allows for more specific parameterizations, like riming of liquid water to falling snow, evaporation and sublimation of rain and snow and supercooled rain. More precipitation types are consequently modeled, now featuring rain, freezing rain, snow, wet snow, rain/snow mixture and ice pellets. Hail and graupel, however, are not modeled. Updates are also included in the




parameterizations of warm-rain autoconversion and accretion, evaporation for small droplets in rain and the ice deposition rate, which is changed to better represent unresolved processes unique for the cloud top environment. Secondary ice production in clouds, such as rime splintering or ice-ice collision break-up, is not modeled.

### 2.2.3 Convection and turbulence

Kinetic energy now dissipates due to convective momentum transport, acting as an additional large-scale heat source. Updraught entrainment is simplified to retain only one process for both turbulent and organized mass exchange and organized detrainment parameterizations are revised. Shallow convection clouds now remain in the liquid phase. Mixing in stable layers is reduced by changing the mixing length and by removing a term in the shear-calculations that accounted for non-resolved shear effects. The boundary-layer scheme is coupled to the cloud scheme through the variance of total water. Cloud ice, however, is now no longer included in the total water distribution.

The momentum roughness length is increased for nine vegetation types and decreased for one, the tundra, following Sandu et al. (2011). Consequently, the average roughness length for momentum is increased, reducing near-surface wind speed, while the roughness length for heat is reduced to account for terrain heterogeneity. Sea ice roughness length for momentum now depends on ice fraction, as sea ice is more likely to break up for partial ice cover, resulting in increased form drag (Andreas et al., 2010). Roughness lengths for heat and momentum for exposed seasonal snow now depend linearly on snow depth, with a roughness length equal to that of local vegetation for snow-free conditions and that of permanent snow for a snow depth of 25 cm or more. For glaciated tiles, however, the roughness lengths for heat and momentum have not changed and still use the expressions developed by Andreas (1987).

### 2.2.4 Surface scheme and lake model

The evaporation formulation for bare soil is revised to improve soil-atmosphere water transfer for sparsely vegetated areas. For ocean tiles, two more layers are activated in IFS cycle 47r1. A cool skin-layer, often only 1 mm thick, is introduced that is typically cooler than the sea surface temperature. This skin layer represents the uppermost layer that loses heat easily by turbulence while shortwave radiation absorption is limited. A warm layer just below the surface of a few meters depth develops during daytime provided wind speeds are low, which is caused by shortwave radiation absorption and induces a diurnal cycle.

The FLake model is included and adds a new surface tile specifically for lakes (Mironov et al., 2010). This replaces the elementary lake model embedded in RACMO since version 2.1 (Van Meijgaard et al., 2008). The surface energy balance (SEB) and skin temperature are therefore determined separately for lakes now. Solar absorption in lakes is allowed below the skin layer using an exponential-decay law. Lakes consist of a top mixed layer with a uniform temperature, and a thermocline below. Ice can form on top of lakes, but fractional ice cover is not allowed and snow on ice is not modeled explicitly. The albedo for lake ice depends on temperature and ranges between 0.4 and 0.7. Lake fraction and depth are fixed fields, the latter is determined from ETOPO1 data (Amante and Eakins, 2009). The addition of a lake surface tile causes thermal inertia, induces phase-change effects and impacts the albedo and roughness lengths (Balsamo et al., 2012).



### 2.2.5 Snow on non-glaciated surfaces

By default in the IFS cycle 47r1, a single-layer model is employed for seasonal snow in non-glaciated terrain. In R24, however, this is replaced by a multi-layer snow module of up to 5 layers (Arduini et al., 2019) that is available as an option within IFS cycle 47r1. For a sufficiently-thick snow pack there are always three layers within 35 cm of the surface. The fourth layer is the accumulation layer, while layer 5 is only up to 15 cm thick and interacts with the soil below. The multi-layer snow scheme includes a new parameterization of heat conductivity, resulting in a more realistic representation of snow density and temperature gradients, while liquid water is allowed to percolate using the tipping-bucket method. Density variations due to blowing snow are allowed, but snow metamorphism is still modeled in a relatively simple way without the formation of complex snow crystals. Snow albedo decays exponentially over time for both dry and wet conditions, but at different rates. In IFS cycle 33r1, the snow albedo recovered whenever snowfall exceeded $1\,\mathrm{mm\,hr^{-1}}$; now it recovers if $10\,\mathrm{kg\,m^{-2}}$ of fresh snow is deposited. Snow fraction now depends linearly on snow depth, with 10 cm of snow for full cover. Previously, only snow mass was taken into account, which resulted typically in a snow depth of 15 cm for full cover, depending on snow density.

### 2.2.6 Climatological data and land-ice mask

The following climatological data are updated as part of IFS cycle 47r1. A new surface elevation field has been implemented based on five data sets: the Shuttle Radar Topography Mission 30 degrees (SRTM30) for 60°N – 60°S, Global Land One-kilometer Base Elevation (GLOBE) north of 60°N, Radarsat Antarctic Mapping Project version 2 (RAMP2) for Antarctica, Byrd Polar Research Center (BPRC) data for Greenland and Iceland Digital Elevation Model for Iceland. For the land-sea mask, GlobCover data (Arino et al., 2007) is used globally except for Antarctica, where RAMP2 is used instead. Monthly Moderate Resolution Imaging Spectroradiometer (MODIS) MOD15A2 satellite data are used for high and low vegetation to determine the leaf area index, instead of a fixed value (Boussetta et al., 2013). Background albedo except for snow is now based on MODIS 5-year climatology split into six components: isotropic, volumetric and geometric for ultra-violet/visible and near-infrared light (Schaaf et al., 2002). The land-ice mask is based on BedMachine Antarctica version 3, at 450 m resolution (Morlighem et al., 2020; Morlighem, 2022a) and BedMachine Greenland version 5, at 150 m resolution (Morlighem et al., 2017; Morlighem, 2022b). The BedMachine products are not part of IFS cycle 47r1 and are regridded to the RACMO grid. For Greenland and the Arctic beyond the domain of BedMachine Greenland, the ice mask introduced in R23p2 is employed when possible. Any remaining ice fields are set using the Global Land Cover Characteristics version 2.0 (GLCC2.0) glacier mask of the IFS (Loveland et al., 2000), but some faulty data have been removed manually.

### 2.2.7 Other changes and tuning

Several more changes are implemented that are not directly related to the updated physical processes of IFS cycle 47r1. Fractional ice cover, with a minimum ice fraction of 0.1 for glaciated grid points, is now allowed, replacing the binary field previously employed. Figure 1 shows the ice masks of the default Greenland 5.5 km, which includes Iceland and Svalbard, and Antarctica 11 km grids. Grid points can therefore be partially covered in ice. In Antarctica, partial ice cover results for instance





in a better representation of the McMurdo Dry Valleys. The Warner-McIntyre-Scinocca non-orographic gravity wave scheme is turned off, as R24 does not resolve the stratosphere sufficiently well for it to work properly. Upper air relaxation, as is described by Van de Berg and Medley (2016), is now applied to moisture as well in order to mitigate accumulation of moisture in the uppermost atmospheric layers in large domains like the Arctic and Antarctic. Some moisture problems at the lateral boundaries

are also fixed and the minimum moisture content for an atmospheric layer is raised to $10^{-6}$ kg kg$^{-1}$. In addition, application of the cloud scheme is extended to the uppermost atmospheric layer, which is not the default in IFS, in order to prevent moisture accumulation. A parameterization has been added to better represent the albedo of superimposed ice in glaciated areas, by linearly interpolating the impurity concentration to be used in the snow albedo scheme between the value of snow and bare ice by applying snow-layer density. This parameterization typically results in a lower albedo for superimposed ice. The background

bare ice albedo field that is converted to an impurity concentration, as described in Van Dalum et al. (2020), has been adjusted to solve interpolation errors previously present in some glaciated grid points close to adjacent non-glaciated land or ocean. This typically results in a lower bare ice albedo for such grid points. A thorough revision of the blowing snow scheme, as is described by Gadde and Van de Berg (2024), is also implemented. These modifications include a better coupling of the atmospheric profile with the blowing snow model and a better representation of the ice particle radius size classes, allowing

snow to be lifted to higher levels in the atmosphere. Sublimation of drifting snow is included in a more realistic fashion, by adding the water vapor to the correct model layers, rather than to the lowest model layer. Furthermore, the refreezing grain size is set to 2 mm for Greenland and the Arctic and 0.4 mm for Antarctica.

The advection of snow hydrometeors leads to unrealistically high precipitation extremes in some coastal mountainous regions, like south-east Greenland and the Antarctic Peninsula. To lower the precipitation extremes, the fall speed of snow

is increased to 2 m s$^{-1}$, resulting in more snow precipitating off-shore. To further reduce the amount of snow hydrometeors precipitating, the riming of falling snow is increased by a factor 2. Forbes and Ahlgrimm (2014) show that uncertainties arise when the IFS cloud parameterizations are used for the Arctic, in particular for mixed-phase clouds. Optimizing parameterizations for one region often leads to biases in others, and as the IFS typically runs on a global domain, compromises are consequently made resulting in parameterizations that are not optimized for polar regions. Furthermore, there are still uncertainties in several

processes related to Arctic clouds, despite recent progress (Kay et al., 2016; Griesche et al., 2021). Therefore, we apply some tuning to cloud-related processes to better represent polar environments. We have increased the Wegener-Bergeron-Findeisen process, responsible for quickly converting mixed-phase clouds into ice clouds by deposition, by a factor of 2.5 by introducing a tunable parameter. The critical autoconversion threshold for snow is decreased and set to $5 \cdot 10^{-7}$ kg kg$^{-1}$, resulting in ice converting to snow more rapidly. To increase the emissivity of clouds and the ablation slightly, longwave downward radiation

is increased by 1%. Some advection parameters in the atmospheric dynamics part of the code are also revised to reduce noise in the semi-Lagrangian integrations of trajectory computations, following findings reported in Table 1 of McDonald (1999).



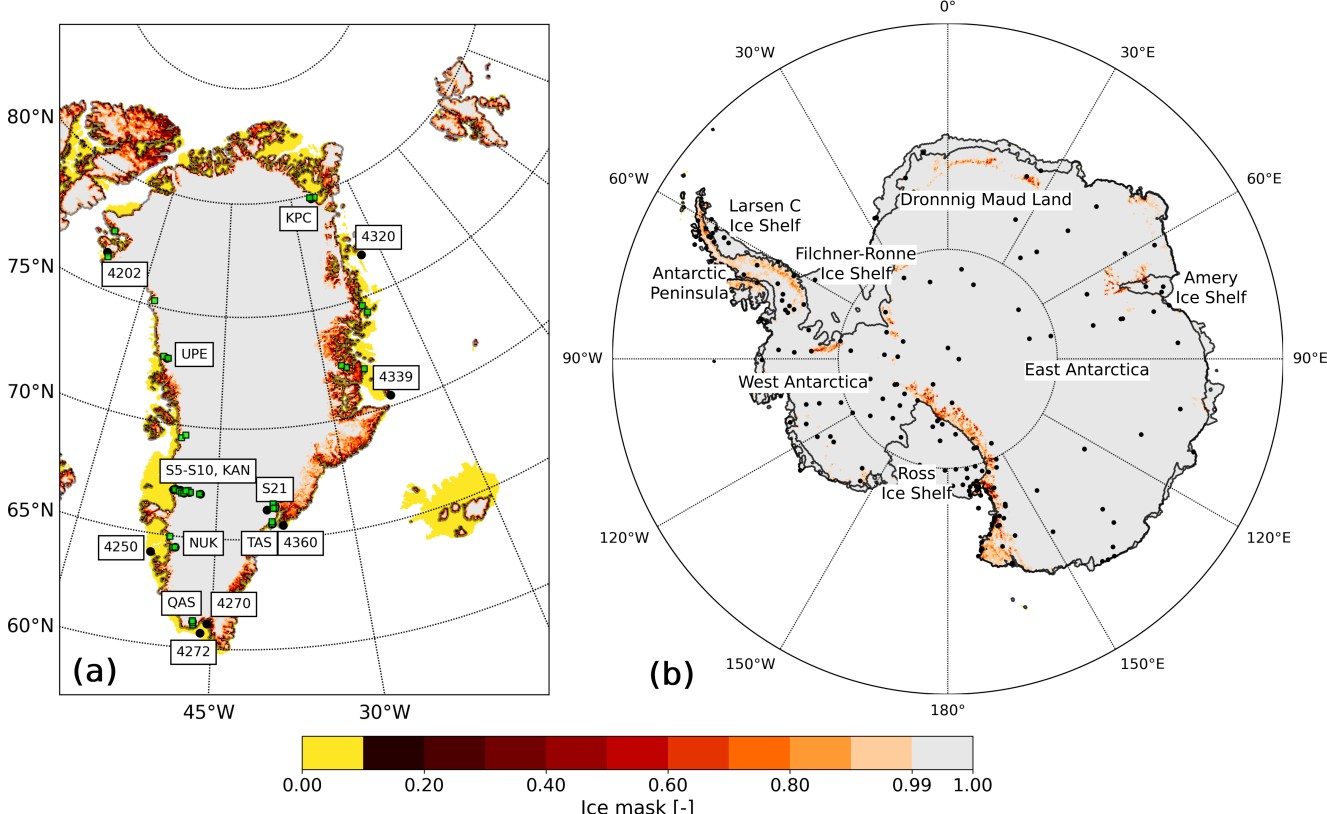

**Figure 1.** Default ice mask of **(a)** Greenland and surroundings, Iceland and Svalbard on a 5.5 km resolution grid, with DMI weather stations (numbered stations), K-transect (S5-S10) and PROMICE locations. Not all PROMICE stations are shown separately due to the close proximity to one another. Green squares illustrate locations of surface mass balance observations used in this study (Machguth et al., 2016). **(b)** Default ice mask of Antarctica on a 11 km resolution grid, including locations of AntAWS stations used in this study. An ice fraction lower than 0.1 is considered ice free (yellow). Note that these ice masks are not used for experiments in this study to enable a direct comparison with the previous model version.

## 3 Methods

### 3.1 RACMO2.4 model simulations

First model results of R24 are shown on three domains: Greenland, the Antarctic and the Arctic. To enable comparison with

previous model versions, the horizontal resolution is reduced from the default high resolution of R24 for the Greenland and Antarctic experiments. The Greenland experiment is conducted on a 11 km grid (default 5.5 km) covering the GrIS, Svalbard, Iceland and parts of the Canadian Arctic and its surrounding areas, and covers 2006 to 2015, with September 2000 to December 2005 as spin up. The Antarctic experiment is carried out on a 27 km grid (default 11 km) and covers 2006 to 2015, with September 2000 to December 2005 as spin up. For both domains, the initial firn-column state is a fully developed firn-column



from R23p3. The ice mask of R23p3 is used (Van Dalum et al., 2020), without fractional ice cover, such that the number of glaciated grid points is the same as in R23p3. Other model settings are set as before, except for the refreezing grain size (Sect. 2.2.7). For the Greenland experiment, R23p3 is forced with 6-hourly ERA-interim data (Dee et al., 2011) at the boundaries, while R24 is forced with 3-hourly ERA5 data (Hersbach et al., 2020). For Antarctica, both model versions are forced with 3-hourly ERA5 data. In all experiments, RACMO is run with 40 atmospheric layers.

For the first time, RACMO is run on a pan-Arctic domain, aligning with the Arctic CORDEX standard (https://climate-cryosphere. org/arctic/, last access: 29-04-2024). The experiments are carried out on a 11 km grid and analyzed during autumn, winter and spring of 2002-2003. For initialization, the R23p3 input file of the Greenland domain is rasterized such that the firn-column of R23p3 is used for as many grid points as possible, while a reference snow pack is generated for grid points without a firn-column, allowing R24 to form a realistic snow pack on new glaciated grid points after spin up. For the Arctic domain, the

refreezing grain size is set to 2 mm.

## 3.2 Surface mass balance and surface energy budget

In RACMO, the residual of the surface energy balance (SEB) contributes to melt $M$:

$$M = \mathrm{SW}_{\mathrm{net}} + \mathrm{LW}_{\mathrm{net}} + \mathrm{LHF} + \mathrm{SHF} + G_{\mathrm{s}}, \tag{1}$$

with $\mathrm{SW}_{\mathrm{net}}$ and $\mathrm{LW}_{\mathrm{net}}$ the net shortwave and longwave radiation absorbed at the surface, respectively. The $\mathrm{SW}_{\mathrm{net}}$ and $\mathrm{LW}_{\mathrm{net}}$ are defined as the sum of the downward and upward flux: $\mathrm{SW}_{\mathrm{net}} = \mathrm{SW}_{\mathrm{d}} + \mathrm{SW}_{\mathrm{u}}$ and $\mathrm{LW}_{\mathrm{net}} = \mathrm{LW}_{\mathrm{d}} + \mathrm{LW}_{\mathrm{u}}$. Fluxes directed away from the surface, like $\mathrm{SW}_{\mathrm{u}}$ and $\mathrm{LW}_{\mathrm{u}}$, have negative values. SHF and LHF are the turbulent sensible and latent heat fluxes and $G_{\mathrm{s}}$ the subsurface heat flux. Part of incoming shortwave radiation does not contribute to the SEB, as some shortwave radiation can be absorbed in subsurface layers by penetrating the surface, causing heating below the surface and induce internal

melt (Van Dalum et al., 2019).

The surface mass balance (SMB) is defined as:

$$\mathrm{SMB} = \mathrm{PR} - \mathrm{RU} - \mathrm{ER} - \mathrm{SU}, \tag{2}$$

with PR precipitation, RU runoff, ER drifting snow erosion and SU sublimation, which includes surface sublimation and

sublimation of suspended blowing snow in the atmosphere. Any excess melt or rain that cannot be retained within the snow pack of a glaciated grid point is modeled as runoff and contributes to the SMB. The SMB therefore also includes mass changes in snow layers near the surface, which is formally referred to as the climatic mass balance (Cogley et al., 2011).





## 3.3 Observational data sets

Several observational data sets are used to evaluate R24. Here, we provide a brief overview.

### 250 3.3.1 Surface mass balance observations in Greenland

The SMB is compared to observations of Machguth et al. (2016) and covers many locations around the margins of the GrIS and surrounding glaciers (locations are shown in Fig. 1a). Most observational sites are located in the ablation zone, but accumulation zone measurements along the Kangerlussuaq-transect (K-transect, Smeets et al., 2018) are also included. Observations predominately consist of stake and snow pit measurements. Only measurement sites that are also located on a 255 glaciated grid point in RACMO are included in this study.

### 3.3.2 Automatic weather station observations from IMAU and PROMICE

The modeled daily-averaged SEB components and near-surface temperature and wind speed are compared with IMAU automatic weather station (AWS) measurements along the K-transect (Smeets et al., 2018) and Programme for Monitoring of the Greenland Ice Sheet (PROMICE, Fausto et al., 2021) on the GrIS. The turbulent heat fluxes are calculated from the single-level 260 observations using the observed surface temperature and a site-specific, temporally varying, surface aerodynamic roughness (Van Tiggelen et al., 2023). Near-surface meteorological quantities are corrected to standard heights using similarity theory flux-gradient relationships, using the stability corrections functions from Holtslag and Bruin (1988). Only the stations that are located on the RACMO 11 km ice mask and were functioning prior to 2016 are considered. This includes the following stations: S5, S6, S9, S10, S21 (IMAU) and KAN_L, KAN_M, KAN_U, QAS_L, QAS_U, QAS_A, TAS_L, TAS_U, TAS_A, 265 KPC_L, KPC_U, UPE_U and NUK_U (PROMICE) (locations are shown in Fig. 1a). Days are discarded when the sensor height above the surface is smaller than 1 m or when at least one hour of measurements are missing.

### 3.3.3 Surface climate Greenland-DMI data collection

The Danish Meteorological Institute (DMI) has produced an extensive observational data set around the GrIS. The Greenland-DMI historical climate data collection (Cappelen, 2021) contains average, maximum and minimum daily temperature, average 270 atmospheric pressure at mean sea level and accumulated precipitation, starting in 1991. Here, we compare all available precipitation measurements with RACMO model output between 2006 and 2015 (locations are shown in Fig. 1a).

### 3.3.4 Antarctic surface melt from QuikSCAT

Melt water fluxes in Antarctica are compared with calibrated QuikSCAT melt product (QSCAT). QSCAT is a melt product that covers the entire AIS and combines in-situ with remote-sensing observations using empirical relations (Trusel et al., 2013). 275 QSCAT data, provided on a 4.45 km resolution grid, is converted to the RACMO grid and compared with RACMO output for the years 2007-2009. As QSCAT provides data around austral summer, the daily melt output of RACMO is summed from mid July to mid July. Here, years refer to the second austral summer year, e.g., 2007 refers to the summer of 2006-2007.



### 3.3.5 Antarctic temperature observations from AntAWS

Wang et al. (2023) have merged observations of air temperature, air pressure, wind speed and direction and relative humidity of
267 Antarctic AWSs, available between 1980 and 2021, into a new dataset, AntAWS. Stations are located across the AIS, with
the highest density in West Antarctica. For the years 2006-2015, 185 AWSs have data and are used in this study for evaluation
(displayed in Fig.1b).

### 3.3.6 Arctic snow depth observations

Snow depth data are from the Copernicus Climate Change Service (C3S) Global Land and Marine Observations Database
(C3S, 2021). This data set includes in-situ observations around the world of many meteorological variables, with most stations
starting around the 1950s. In this study, daily snow depth data within the Arctic model domain between September 2002 and
May 2003 are used, with a minimum of ten days of valid data in a season.

## 4   First results: Greenland

In this section, we evaluate R24 for Greenland and its surroundings and investigate the impact that implemented changes have
on several aspects of the climate system.

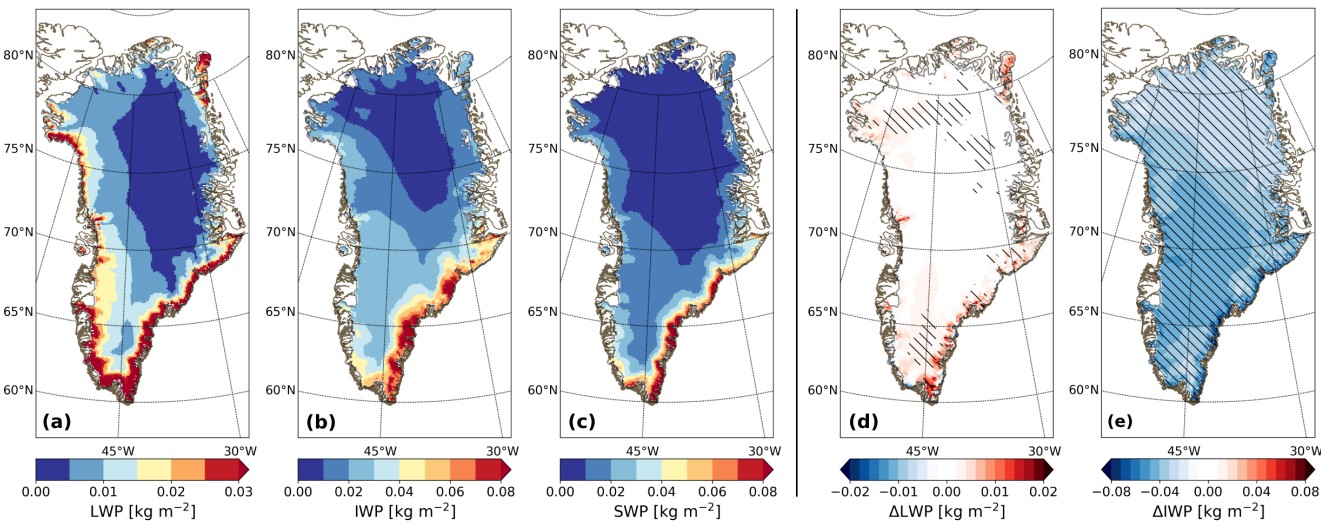

**Figure 2.** Annual-average vertically-integrated **(a)** liquid water (LWP), **(b)** ice water (IWP) and **(c)** snow content (SWP) for R24, and **(d)**
LWP and **(e)** IWP difference (R24 - R23p3) over the GrIS for 2006-2015. Differences larger than inter-annual variability are hatched. The
color scales of **(a)** and **(b)** are similar to Van Tricht et al. (2016).





## 4.1 Cloud content

Figures 2a and b show the vertically-integrated liquid (LWP) and ice water content (IWP) of R24, respectively. Comparing
Fig. 2a with b illustrates that cloud content predominantly consist of ice. Typically, however, liquid and ice water coexist
in mixed-phase clouds. The south-eastern margin is characterized by high cloud content, while it diminishes northward and
towards the interior. A similar pattern is modeled for LWP in R23p3, but the IWP is considerably higher (Figs. 2d and e).
The pattern and values of LWP and IWP of R24 are similar to results of Van Tricht et al. (2016), which are based on satellite
observations. The domain-averaged values for R24 of IWP and LWP are 0.022 and 0.010 kg m$^{-2}$, respectively, and 0.060
and 0.009 kg m$^{-2}$ for R23p3, while Van Tricht et al. (2016) report 0.021 and 0.009 kg m$^{-2}$. Compared to Van Tricht et al.
(2016), the IWP of R23p3 is overestimated, which has improved in R24. Note that the Cloud-Aerosol Lidar with Orthogonal
Polarization (CALIOP) sensor on-board of the Cloud-Aerosol Lidar and Infrared Pathfinder Satellite Observation (CALIPSO)
satellite, which is designed to observe optically-thin clouds and is used in the study of Van Tricht et al. (2016), has difficulties
observing clouds close to the surface if higher-elevated optically-thick clouds obstruct lidar backscatter, and if surface scatter
contaminates radar imagery (Kay et al., 2008).

Contrary to R23p3, R24 explicitly determines rain and snow hydrometeors, i.e., rain and snow that are suspended in the
atmosphere or are precipitating, but have not yet reached the surface. Rain hydrometeor content (not shown) is typically low all
around the GrIS except for small areas in the south. Snow hydrometeor content, on the other hand, is more pronounced (Fig.
2c), following a similar pattern as IWP (Fig. 2b), and is particularly high in mountainous terrain in the south-east.

## 4.2 Precipitation

Figure 3a shows average annual precipitation between 2006 and 2015 for R24. Large amounts of precipitation, locally more
than 3000 mm water equivalent (w.e.) yr$^{-1}$, are modeled in coastal regions of the south-east that are characterized by steep
slopes. Strong moisture influx associated with the North Atlantic storm tracks and Icelandic Low therefore results in strong
orographic lift and strong precipitation in this region (Berdahl et al., 2018). Further in-land and northward, precipitation is
lower. Compared to R23p3, R24 models 7.8% more precipitation on the GrIS: 790 Gt yr$^{-1}$ versus 733 Gt yr$^{-1}$ (Table 1). In
the GrIS interior, precipitation is virtually the same as in R23p3 (Fig. 3b), while it is generally higher around the margins,
especially in the south-east (region **A**, Fig. 3b), where precipitation has significantly increased, locally by more than 500 mm
w.e. yr$^{-1}$. Close to the ice sheet on adjacent lower-elevated tundra and coastal waters, however, precipitation has decreased
locally.

To understand these patterns, Fig. 3c shows the average net snow-hydrometeor mass loss as is calculated in the cloud scheme
of the IFS physics package in R24. Because there is no horizontal transport of snow hydrometeors between grid cells within the
cloud scheme, positive values in Fig. 3c indicate that on average more snow-hydrometeor mass is lost than is produced locally.
This is only possible if hydrometeor content is increased by horizontal transport from nearby grid cells before precipitation is
calculated in the cloud scheme. This process is now represented in R24, as hydrometeors are treated as prognostic variables
permitting horizontal transport between neighboring grid cells. Figure 3c therefore essentially illustrates the net effect of





**Figure 3. (a)** Annual-average precipitation for 2006-2015 for R24, with stepsizes of 100, 250 and 500 mm w.e. yr$^{-1}$ between 0 to 500, 250 to 500, and 1500 to 3000 mm w.e. yr$^{-1}$, respectively. Elevation contours are shown in black, with a 500 m interval. **(b)** Precipitation difference (R24 - R23p3), with a stepsize of 100 and 20 mm w.e. yr$^{-1}$ between -500 to -100 and 100 to 500, and -100 to 100 mm w.e. yr$^{-1}$, respectively. Differences larger than inter-annual variability are hatched. **(c)** Average of net snow-hydrometeor mass loss as is calculated in the cloud scheme of IFS physics in R24. Red indicates a mass loss, blue a mass gain. The same color distribution is used as in (b). Regions **A**, **B** and **C** are discussed in the text.

horizontal transport of snow hydrometeors on modeled precipitation. As hydrometeors are not explicitly modeled in R23p3,
horizontal transport cannot occur and any snow that is formed in R23p3 precipitates to the surface instantaneously in the
same grid cell. The advection of hydrometeors does in principle not change mass across the model domain, but leads to a
redistribution of mass within it. It brings snow to the ice sheet that formed upwind, changing the SMB of the GrIS. Figure 3c
shows that some snow that would have precipitated off the coast or on the lower-lying tundra in R23p3, is now moved onto the



GrIS. This process is especially prevalent around region **A**, where vast amounts of moisture are advected to the GrIS (Berdahl
et al., 2018). Comparing Fig. 3c to region **A** of Fig. 3b illustrates that a large part of this signal is due to the advection of snow
hydrometeors, explaining most of the precipitation difference between R24 and R23p3. For region **B**, snowfall has increased on
steep slopes, but contrary to region **A**, precipitation has decreased somewhat on adjacent lower-lying ice streams. Advection of
snow hydrometeors also impacts this region (Fig. 3c), as hydrometeors formed above ice streams are moved to adjacent slopes,
where they precipitate. Local geometry therefore has a stronger impact on precipitation than in R23p3. Contrary to region **A**,
most snow that is redistributed in R24 for region **B** is formed locally above the GrIS, and total precipitation is hence similar to
R23p3.

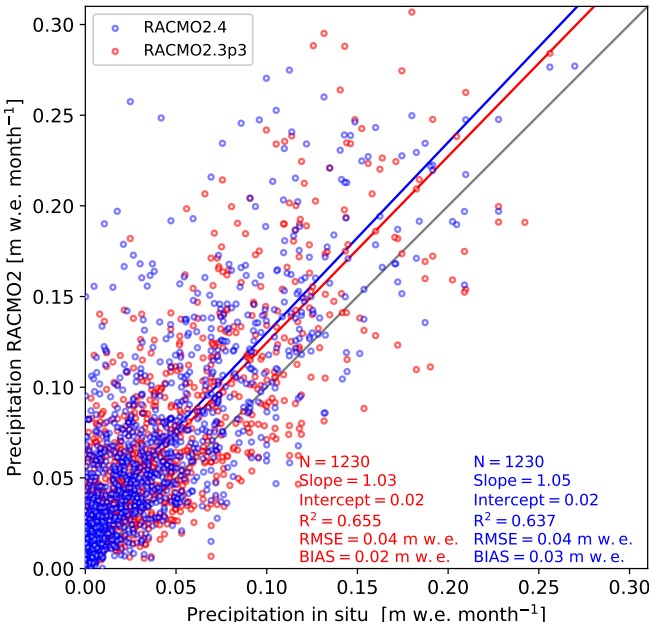

**Figure 4.** Monthly precipitation (m w.e. month$^{-1}$) derived from R24 (blue) and R23p3 (red) compared to the Greenland-DMI historical
climate data for the period 2006-2015. The black line represents the 1-on-1 line. The blue and red line illustrate linear regression of the data.
Slope and intercept of linear regression are displayed, as well as the bias, root-mean-square error (RMSE), determination coefficient (R$^2$)
and number of points (N). Location of observational sites are illustrated in Fig. 1a.

Rainfall on the GrIS is typically small compared to snowfall, except for region **C** and some coastal areas in region **A**. Due
to the high fall speed of rain (usually several meters per second, following the parameterization of Kessler (1969)) and its
formation that often occurs in lower atmospheric layers, advection of rain hydrometeors is negligible at the employed model
resolution and its impact on the GrIS is small. Figure 3b also shows precipitation changes on the Atlantic Ocean, which can
be attributed to micro-physical changes to the cloud scheme. Furthermore, changes in sea-ice extent may impact precipitation
formation. With the addition of advection of hydrometeors, the modeled precipitation patterns of R24 are qualitatively closer





to those of MAR (Fettweis et al., 2020), which already explicitly models advection of hydrometeors since several model iterations.

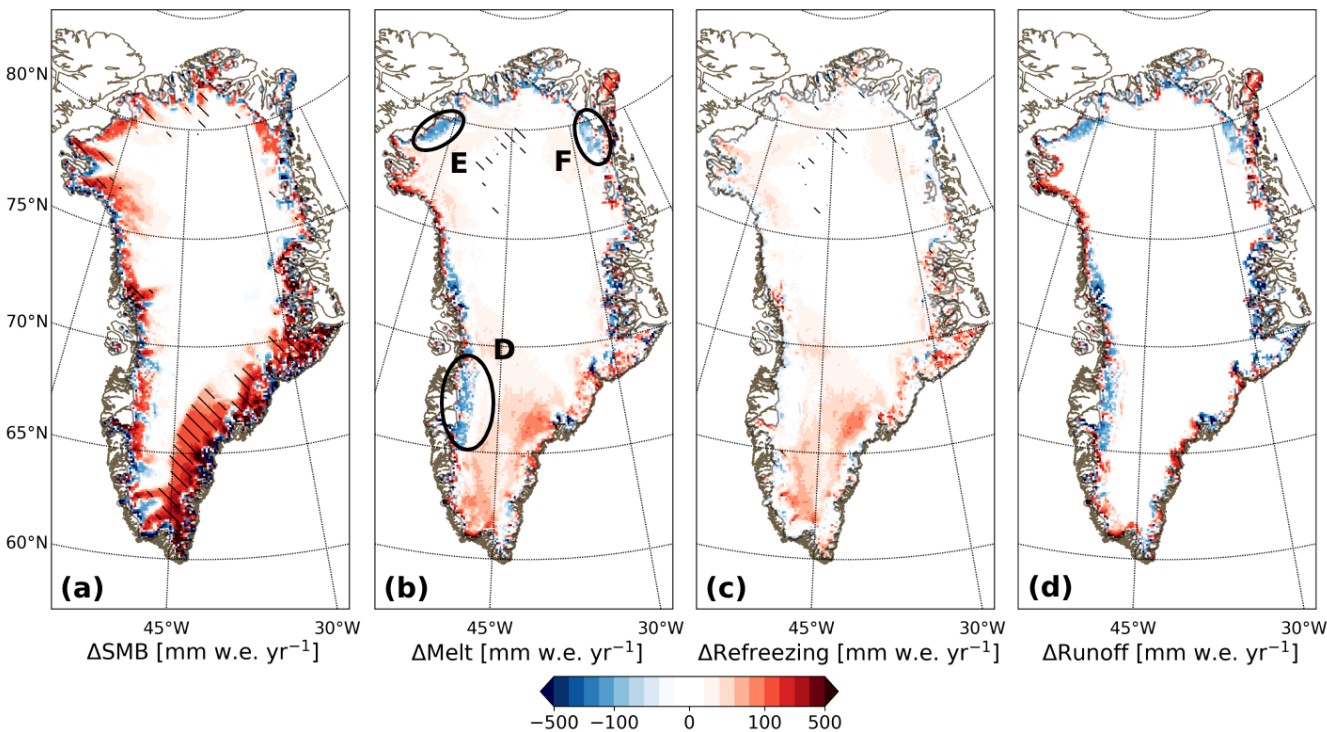

**Figure 5.** Average (2006-2015) difference (R24 - R23p3) for **(a)** SMB, **(b)** melt, **(c)** refreezing and **(d)** runoff in mm w.e. yr$^{-1}$. The same color scale as in Fig. 3b is applied. Differences larger than inter-annual variability are hatched. Regions **D**, **E** and **F** are discussed in the text.

RACMO compares well with precipitation observations of the Greenland-DMI historical climate data collection (Cappelen, 2021), located around the GrIS (locations are shown in Fig. 1a). The differences between R24 and R23p3 are small, but both illustrate, on average, an overestimation with respect to observations, with a bias of 0.03 and 0.02 m w.e., respectively (Fig. 4). The spread in the data increases with precipitation, illustrating that more uncertainty is associated with wetter conditions. Differences between R24 and R23p3 are more substantial locally. For station 4202 (located near Pituffik) and 4320 (located

near Danmarkshavn) located in the dry northwest and northeast, the bias is comparable to R23p3. A small improvement is observed at station 4272 (located near Qaqortoq), where rain often occurs. The bias difference with respect to R23p3 of nearby station 4270 (located near Narsarsuaq), however, is negligible. This illustrates that the resolution of RACMO, especially when it is reduced to 11 km, is not high enough the capture the impact of local effects on precipitation. Excess precipitation is now modeled at the other stations, especially at station 4339 (located near Ittoqqortoormiit) that is located close to orography.



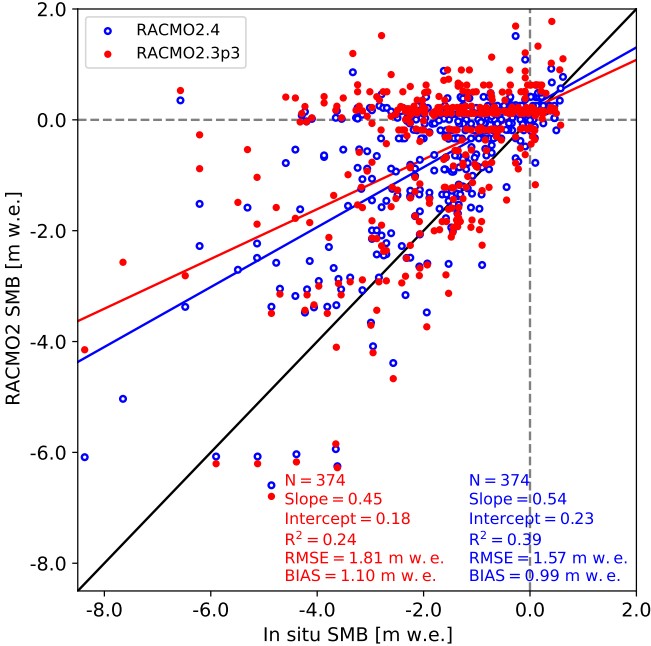

**Figure 6.** Surface mass balance for 2006-2015 for R24 (blue) and R23p3 (red) with respect to in situ observations of Machguth et al. (2016). The 1-on-1 line is shown in black. Bias, root-mean-square error (RMSE), determination coefficient ($R^2$), intercept and slope of linear regression, and number of points (N) are also shown. Locations are shown in Fig. 1a.

## 4.3 Surface mass balance

The SMB difference between R24 and R23p3 is shown in Fig. 5a. The figure shows that the SMB signal is dominated by precipitation changes (Fig. 3b) for most regions, in particular for regions **A**, **B** and **C**. Despite increased precipitation, snow melt has increased in the interior of the southern GrIS (Fig. 5b). As the firn layer is not fully saturated there, all melt refreezes (Fig. 5c) and no additional runoff is modeled (Fig. 5d). This melt increase is associated with changes in the snow structure; in particular increased refreezing grain size that leads to more absorption of solar radiation. Closer to the ice-sheet margin where firn becomes saturated or is absent, this process leads to more runoff, which occurs in particular in regions **A** and **C**. In addition, a reduction in snowfall close to the coast in region **A** results in exposure of bare ice earlier in the season, enhancing melt and runoff, while for region **C**, an increase in rain augments melt and runoff. The ablation zone of south-west Greenland around the K-transect (region **D**, Fig. 5b) shows less melt and runoff, resulting in an SMB increase. This is induced by a precipitation increase (Fig. 3b) that thickens the snow layer, delaying the exposure of bare ice typically by one or two weeks, shortening the ablation season. Similarly, a precipitation increase delays exposure of bare ice in regions **E** and **F**. As these regions are characterized by dry conditions (Fig. 3a), the small precipitation increase shown in Fig. 3b is, by percentage, roughly 10 to 30%. Consequently, the snow pack is a few centimeters thicker at the start of the ablation season and therefore takes longer to melt, exposing bare ice later and reducing melt and runoff. Around the margins of the GrIS, some outermost grid points that



**Table 1.** Ice-sheet-integrated surface mass balance (SMB) and its components in Gt yr$^{-1}$: precipitation (PR), melt (ME), refreezing (RF), runoff (RU), drifting snow erosion (ER) and sublimation (SU) for Greenland (GRL) and Antarctica (ANT) for 2006-2015, for R24, R23p3, R24 - R23p3 ($\Delta$R23p3), the standard deviation ($\sigma$) for R24 and R23p3. For R24, SU includes surface sublimation and sublimation of snow in suspension in the atmosphere due to blowing snow.

| | GRL | | | | | ANT | | | | |
| --- | --- | --- | --- | --- | --- | --- | --- | --- | --- | --- |
| Variable | R24 | R23p3 | $\Delta$R23p3 | $\sigma_{R24}$ | $\sigma_{R23p3}$ | R24 | R23p3 | $\Delta$R23p3 | $\sigma_{R24}$ | $\sigma_{R23p3}$ |
| SMB | 404.4 | 341.4 | 63.0 | 82.3 | 70.7 | 2513.4 | 2530.4 | -17.0 | 83.9 | 73.5 |
| PR | 789.5 | 732.6 | 56.9 | 64.9 | 56.3 | 2767.0 | 2711.7 | 55.3 | 79.4 | 71.8 |
| ME | 631.6 | 609.5 | 22.1 | 145.6 | 129.7 | 123.5 | 109.8 | 13.8 | 22.5 | 23.0 |
| RF | 314.9 | 285.9 | 29.0 | 79.1 | 65.5 | 124.4 | 108.9 | 15.5 | 25.1 | 24.8 |
| RU | 351.7 | 356.0 | -4.3 | 84.0 | 80.3 | 2.9 | 4.9 | -2.0 | 1.6 | 2.9 |
| ER | 0.9 | 1.0 | -0.1 | 0.2 | 0.0 | 9.4 | 5.5 | 3.9 | 0.4 | 0.2 |
| SU | 32.5 | 34.2 | -1.7 | 3.3 | 1.7 | 247.7 | 170.9 | 76.8 | 9.2 | 6.4 |

are adjacent to non-glaciated land or ocean show enhanced melt leading to an SMB reduction. This is induced by modifications in the bare ice albedo field that is coupled to RACMO (described in Sect. 2.2.7), typically resulting in a lower albedo and more melt.

Compared to in situ observations of Machguth et al. (2016), R24 and R23p3 show a similar pattern, but the SMB is overestimated for most locations (Fig. 6). In particular for locations that are close to the ice sheet margin and are characterized

by strong ablation, such as S5 or QAS_L, is the SMB too high and not enough runoff is modeled in RACMO. Melt is typically underestimated at such stations, as the SHF is on average too low (Noël et al., 2018). These stations are also impacted by significant winter snow distribution heterogeneity, where winter snow collects in gullies while the measurements are often performed on snowfree hummocks, which are not represented in the model (see Fig. 4 of Van den Broeke et al. (2008)). For measurement sites that are located at a higher elevation, such S6 or S7, the SMB is closer to observations, where differences

between R24 and R23p3 are dominated by precipitation changes. The bias, root-mean-square error (RMSE) and determination coefficient of R24 have improved compared to R23p3, as the SMB of R24 is typically lower than R23p3 and closer to observations, in particular in high-ablation areas. Some measurements are done in close proximity to each other and the horizontal resolution of RACMO is insufficient to properly resolve them, resulting in the same RACMO grid point used for several in situ locations.

Integrated over the ice sheet (Table 1), the SMB of R24 has increased by 63.0 Gt yr$^{-1}$ with respect to R23p3, which is slightly lower than the standard deviation of R23p3 (70.7 Gt yr$^{-1}$). This difference is mostly caused by a precipitation increase (56.9 Gt yr$^{-1}$), which is in agreement with Fig. 5b. Runoff has decreased slightly (4.3 Gt yr$^{-1}$). Out of the 32.5 Gt yr$^{-1}$ total (surface and blowing snow) sublimation that occurs in R24, 24.5 Gt yr$^{-1}$ occurs on drifting snow. Total sublimation is marginally decreased (1.7 Gt yr$^{-1}$), despite more efficient sublimation due to blowing snow, as the modifications in the blowing

snow scheme of Gadde and Van de Berg (2024) generally result in less vigorous blowing snow on the GrIS. This manuscript





**Figure 7.** Daily-averaged surface energy balance components, temperature and wind speed for R24 with respect to PROMICE and IMAU AWS observations on the GrIS for 2006-2015, for **(a)** shortwave downward (SWd) and **(b)** upward (SWu) radiation, **(c)** longwave downward (LWd) and **(d)** upward (LWu) radiation, **(e)** sensible heat flux (SHF), **(f)** latent heat flux (LHF), **(g)** 2-m temperature (T2m) and **(h)** 10-m wind speed (w10m). The 1-on-1 line is shown in black. Bias, root-mean-square error (RMSE), determination coefficient ($R^2$), intercept and slope of orthogonal total least squares regression, and number of points (N) are also shown.

does not present a new surface mass balance product; this follows in a later publication, where a more extensive SMB analysis will be done.





**Table 2.** RMSE, bias and determination coefficient ($R^2$) of R24 and R23p3 SEB components, temperature and wind speed output with respect to IMAU and PROMICE AWS observations in Greenland for 2006-2015, for shortwave downward (SWd) and upward (SWu) radiation, longwave downward (LWd) and upward (LWu) radiation, sensible heat flux (SHF), latent heat flux (LHF), 2-m temperature (T2m) and 10-m wind speed (w10m).

| | | RMSE | | Bias | | $R^2$ | |
|---|---|---|---|---|---|---|---|
| Variable | Unit | R24 | R23p3 | R24 | R23p3 | R24 | R23p3 |
| SWd | W m$^{-2}$ | 24.5 | 27.8 | -0.60 | 2.68 | 0.99 | 0.99 |
| SWu | W m$^{-2}$ | 28.1 | 34.2 | -4.47 | -8.87 | 0.98 | 0.97 |
| LWd | W m$^{-2}$ | 24.3 | 19.9 | -14.5 | -5.47 | 0.96 | 0.96 |
| LWu | W m$^{-2}$ | 11.4 | 10.1 | 6.56 | 2.59 | 0.99 | 0.99 |
| SHF | W m$^{-2}$ | 19.0 | 20.8 | 0.28 | -3.73 | 0.86 | 0.83 |
| LHF | W m$^{-2}$ | 10.5 | 9.97 | 4.84 | 4.15 | 0.85 | 0.85 |
| T2m | °C | 2.18 | 2.23 | -0.63 | 0.27 | 0.99 | 0.99 |
| w10m | m s$^{-1}$ | 1.77 | 2.09 | -0.39 | -0.97 | 0.94 | 0.93 |

## 4.4 Surface energy balance, temperature and wind speed

Figure 7 shows the SEB components, 2-m temperature (T2m) and 10-m wind speed (w10m) for R24 with respect to IMAU
and PROMICE AWS observations. The RMSEs, biases and determination coefficients of R24 and R23p3 are also summarized in Table 2.

Compared to observations, the shortwave (SW) radiation of R24 fits well (Fig. 7a and b), with a bias of -0.60 W m$^{-2}$ for $SW_d$ and -4.47 W m$^{-2}$ for $SW_u$. Despite the larger spread in $SW_d$ radiation between 150 and 350 W m$^{-2}$, which represents cloudy conditions, the total-least square fit is close to the 1-on-1 line, indicating that mismatches between model and observations
average out. The good agreement between both upward and downward SW radiation illustrates that the albedo is, on average, also modeled well. This is important, as $SW_{net}$ is the main energy source for melt.

Figure 7c shows that R24 underestimates downward longwave radiation (LWd), in particular for lower values (between 100 and 250 W m$^{-2}$) that are typically associated with clear-sky conditions, thin clouds and/or low temperatures. The T2m of R24 fits generally well with observations (Fig. 7g), with a small bias (-0.63°C). Consequently, the upward longwave radiation
(LWu) is also underestimated somewhat (less negative, Fig. 7d). As the T2m bias is small and upper air temperatures are tied to ERA5, the LWd differences with respect to observations are likely caused by biases in cloud cover, atmospheric aerosols and/or LW emissions of clouds in dry and cold conditions.

The bias and RMSE of the turbulent fluxes are small (Fig. 7e and f). In particular, the sensible heat flux (SHF) is modeled well on average, with positive values for a flux towards the surface, but local differences can be larger. For example, if observations
from S5 are only considered, the SHF bias (-16.4 W m$^{-2}$) and RMSE (25.7 W m$^{-2}$) are larger. Low-lying ablation stations like S5 and QAS_L are characterized by rough terrain and a horizontal resolution of 11 km is inadequate to capture this properly





**Figure 8.** Average (2006-2015) difference (R24 - R23p3) for **(a)** SMB, **(b)** precipitation and **(c)** melt. **(d)** Annual-average melt difference with QSCAT (R24 - QSCAT) for 2007-2009. Differences larger than inter-annual variability are hatched.



(Fausto et al., 2016; Van de Berg et al., 2020), resulting in too weak turbulent exchange in R24, especially when a high SHF is observed. Similarly, the latent heat flux (LHF) is modeled well, but the RMSE (19.9 W m$^{-2}$) and bias (14.2 W m$^{-2}$) are larger for KAN_L, which is located close to S5. A misrepresentation of wind speed may also lead to enhanced or weakened turbulent

mixing. On average, however, the wind speed in R24 is modeled well (Fig. 7h). Similarly to Fig. 7, all SEB components, T2m and w10m modeled by R23p3 are shown in Fig. A1.

## 5 First results: Antarctica

### 5.1 Surface mass balance

Most of the AIS can be characterized as a polar desert, in particular on the East Antarctic plateau, where less than 50 mm w.e.

yr$^{-1}$ of precipitation is common (Nicola et al., 2023). However, due to the sheer size of Antarctica, the total annual precipitation is high (Table 1, 2767 Gt yr$^{-1}$), while runoff is negligible in R24. The only major mass loss term is sublimation (248 Gt yr$^{-1}$), of which 222 Gt yr$^{-1}$ occurs on blowing snow. Still, sublimation is an order of magnitude lower than precipitation and relatively constant from year to year. Consequently, the SMB signal and its variability in time and space is dominated by precipitation. With respect to R23p3, the SMB has decreased slightly (17 Gt yr$^{-1}$), which is considerably lower than interannual variability

(73.5 Gt yr$^{-1}$ standard deviation for R23p3), while precipitation has increased by 55 Gt yr$^{-1}$. This is compensated by increased sublimation (77 Gt yr$^{-1}$) due to modifications in the blowing snow scheme (Gadde and Van de Berg, 2024) that result in snow being lifted to higher levels in the atmosphere, reaching unsaturated air above the surface layer, enhancing sublimation. Melt has also increased somewhat, but does not lead to additional runoff as it is compensated by more refreezing.

Figure 8a shows the spatial pattern of SMB changes. An alternating pattern of increased and decreased SMB is visible around

the margins of the AIS, in particular in East Antarctica. Similar to the pattern in region **B** in Greenland (Fig. 3b), advection of snow hydrometeors causes precipitation to increase on slopes while decreasing on adjacent lower-lying ice streams (Fig. 8b). And similar to region **A** in Greenland (Fig. 3b), dominant westerly winds cause horizontal advection of snow hydrometeors that are formed above the ocean but precipitate in the mountains of the Antarctic Peninsula, resulting in significantly increased precipitation with respect to R23p3. On the Filchner-Ronne ice shelf, the SMB has increased somewhat, which is related to

decreased sublimation. A similar sublimation decrease occurs on the Ross ice shelf, but is here compensated by decreased precipitation, resulting in a near-zero SMB change. Compared to R23p3, melt has increased on ice shelves in East Antarctica and the Antarctic Peninsula (Fig. 8c). Melt in R24 compares well with respect to QSCAT (Fig. 8d). QSCAT shows that melt only occurs around the margins of the AIS. On the Filchner-Ronne and Ross ice shelves, melt is limited, which is captured properly by R24. On the other ice shelves in West Antarctica and the Antarctic Peninsula, melt is somewhat underestimated in

R24. For Larsen C, however, melt is overestimated except near the grounding line, while Van Dalum et al. (2022) showed that in R23p3 melt is significantly underestimated for large parts of Larsen C.







**Figure 9. (a)** Annual-average T2m difference (R24 - R23p3) for 2006-2015. Differences larger than inter-annual variability are hatched. **(b)** Daily-average T2m for R24 with respect to AntAWS measurements (locations are shown in Fig. 1b). The 1-on-1 line is shown in gray. Bias, root-mean-square error (RMSE), determination coefficient ($R^2$), intercept and slope of orthogonal total least squares regression, and number of points (N) are also shown. **(c)** RMSE and **(d)** bias for R24 and R23p3 with respect to AntAWS for T2m, with 5 °C bins.





## 5.2 Temperature

Over the AIS, the T2m of R24 is on average lower than R23p3 (Fig. 9a). In particular the western part and the interior of the eastern AIS show significantly lower temperatures. The margins of the eastern AIS, however, are typically characterized by higher temperatures, in particular in Dronning Maud Land and the Amery ice shelf.

R24 compares well to T2m observations (Fig. 9b) of the AntAWS observational data set (Wang et al., 2023), with stations being situated all over the AIS (locations shown in Fig. 1b). The determination coefficient is high and the fit line is close to the 1-on-1 line, in particular for cold conditions (T2m lower than -40 °C). For warmer conditions, the T2m is on average lower than observed. The spread in the data increases between -40 and -10 °C, resulting in a relatively high RMSE (Fig. 9c) compared to colder and warmer conditions. Between -80 to -40 °C, the RMSE has improved compared to R23p3. Between -25 and 0 °C, however, the RMSE has increased in R24 with respect to R23p3. Figure 9d shows the bias for temperature bins, confirming that the temperature is close to observations between -80 and -40 °C, with a small positive bias for most bins, while the temperature is underestimated for temperatures higher than -35 °C. Compared to R23p3, cold conditions have improved, while the bias, to a slightly lesser extent, increased for warmer conditions. The biases over all data are in -1.85 °C for R24 and -0.78 °C for R23p3.

## 6  Snow depth in the Arctic domain

We illustrate the capabilities of R24 to model the polar climate outside of ice sheets and glaciers by analyzing modeled snow depth in a new pan-Arctic domain. Figure 10 shows the snow depth bias for 2002-2003 compared to in-situ observations for autumn, winter and spring (Figs. 10a-c, respectively), with results for Scandinavia enlarged in Figs. 10d-f. In this analysis, observations located in the boundary relaxation zone of R24 are excluded. During autumn, the bias with respect to observations is small, especially in relatively flat terrain like Siberia and Canada (Fig. 10a). In mountainous areas with steep slopes, in particular in Norway, a larger snow depth is modeled than observed (Fig. 10d). Stations close to the coast, however, have again a small bias. Moving further in-land into flatter terrain (i.e., Sweden), the bias becomes smaller. During winter, this signal amplifies with larger biases in steep terrain while the biases remain low in Sweden (Fig. 10b and e), where it is typically within 20% of observed snow depth. A similar positive bias emerges along mountainous terrain of the Pacific coast of North America. In central Eurasia, a negative bias appears, suggesting that the modeled snow pack is not thick enough. During spring, the aforementioned signals persist (Fig. 10c and f), in particular in Norway and central Eurasia.

Arduini et al. (2019) present an analysis of the multi-layer snow scheme that is now also applied in R24. They indicate that this scheme generally improves the bias with respect to the previously employed single-layer scheme. For central Eurasia, Arduini et al. (2019) also report that snow depth is underestimated, possibly due to overestimated compaction or melt in forested areas or due to too little snowfall. They also show a positive bias in mountainous terrain in Norway and the Pacific coast of North America. It seems that these biases are inherited by R24.

Other factors can also contribute to the aforementioned biases. As indicated in Figs. 3b and 8b, precipitation typically has increased for regions with a steep topography, and likely contributes to the observed signal. A misrepresentation of



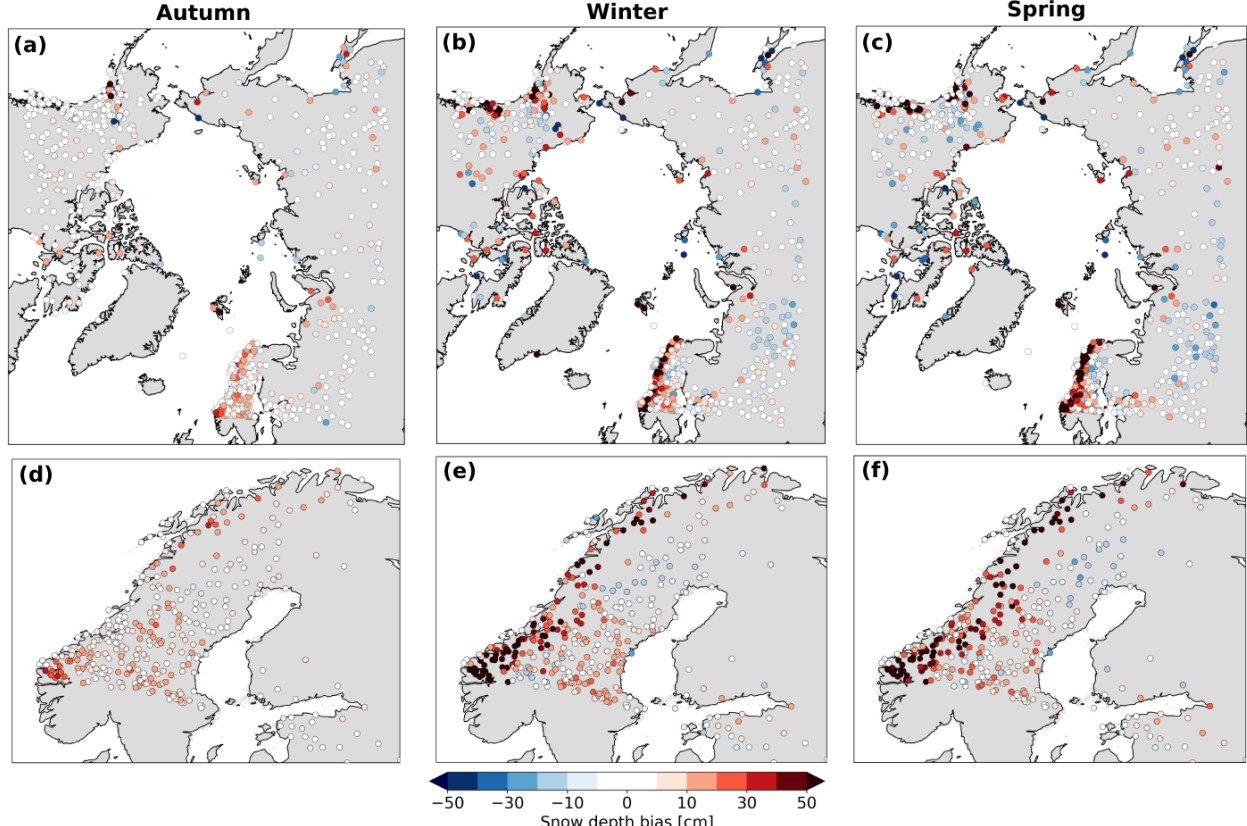

**Figure 10.** Daily snow depth bias of R24 with respect to in-situ observations for 2002-2003 for autumn **(a)**, winter **(b)** and spring **(c)**, with stepsizes of 10 and 5 cm between -50 to -10 and 10 to 50, and -10 to 10 cm, respectively. Scandinavia is shown separately in **(d)-(f)**. Red illustrates an overestimated snow depth in R24. Data points in the boundary relaxation zone (24 grid cells wide) are excluded from the analysis.

475 the orography employed in R24 or too low resolution to properly represent mountainous terrain are also possible partial explanations. Furthermore, as stations are often located in valleys with typically lower amounts of snowfall than the surrounding mountains, the closest R24 grid point may not be representative for such a station. Blowing snow, inhomogeneous snow cover or nonuniform melting may impact the observations and comparison with R24 as well. Furthermore, shading effects of a canopy above snow are not modeled.



## 7   Summary and conclusions

In this study, we presented a new version of the polar regional atmospheric climate model RACMO, referred to as RACMO2.4p1, and provided an overview of all implemented changes. As a proof of concept, we have shown the first model evaluations for Greenland, Antarctica and a new Arctic domain, by comparing it with the previous RACMO iteration, R23p3, and by comparing it with remote sensing and in-situ observations.

The IFS physics module has been updated in R24 from cycle 33r1 to cycle 47r1 and its coupling to the HIRLAM dynamical core is recoded to facilitate better readiblity and more flexibility. IFS cycle 47r1 includes changes in several aspects. More prognostic variables are introduced in the cloud scheme, where cloud ice and water are now treated separately. This results in a better representation of mixed-phase clouds, supersaturation in clouds and super-cooled liquid water, allowing for more specialized parameterizations. In addition, rain and snow hydrometeors are now prognostic variables. More precipitation types are therefore modeled and rain and snow hydrometeors can now be transported horizontally by advection. Eleven CAMS aerosol types are now used, replacing the previous six aerosol types. IFS cycle 47r1 also includes several revisions of the convection and turbulence routines. Furthermore, a new lake model and a multi-layer snow model for non-glaciated tiles are introduced. The stand-alone version of IFS radiation physics, ecRad, is also implemented and includes new parameterizations. Another important addition with respect to R23p3 is a fractional land-ice mask, allowing grid points to be partially covered by ice. Several climatological fields are also updated, parameterizations for glaciated grid points are altered and some tuning is implemented.

For the Greenland ice sheet, cloud water and ice content compare well with satellite observations. Precipitation has changed notably in several regions, and has increased in particular in mountainous regions of the south-east, which can be mainly ascribed to horizontal advection of snow hydrometeors. Integrated over the ice sheet, precipitation has increased, resulting in a higher SMB compared to the previous model version R23p3. Compared to in-situ observations along the margins of the Greenland ice sheet, the SMB shows good agreement, with some improvements with respect to R23p3. Along the K-transect and at PROMICE locations, temperature, wind speed and SEB compare generally well with measurements. There is, however, still room for improvements, for instance in downward longwave radiation, which is now underestimated. For simulations of the Greenland ice sheet, future model development should therefore focus on improving clouds and aerosols and their impact on radiation.

Similar processes cause precipitation changes in Antarctica, in particular for the Antarctic Peninsula, where more precipitation has increased the SMB. Some melt is modeled on the ice shelves, which compares well with QSCAT observations. Integrated over the ice sheet, the SMB and its components are similar to R23p3. T2m compares well with AntAWS data, with better temperature estimates for low temperatures (lower than -40 °C) compared to R23p3. However, R24 underestimates temperatures for warmer conditions (higher than -40 °C).

RACMO is used for the first time on a pan-Arctic domain. Here, we analyzed snow depth by comparing with in-situ observations. In flat terrain, snow depth compares well, but snow depth is underestimated in central Eurasia. In mountainous



regions, R24 has a tendency to overestimate snow depth. A more in-depth analysis for this domain will follow in a later publication.

In conclusion, we have shown that the newly-implemented parameterizations in R24 lead to promising results for Greenland, Antarctica and the Arctic. Precipitation changes have the strongest impact on the SMB, but the SMB compares well with observations nonetheless. Likewise, temperature, SEB and snow depth show promising results. This work therefore lays the foundation for future work with R24. In forthcoming publications, a new near-surface climate and SMB product using R24 will be presented for Greenland and Antarctica.





## 520 Appendix A: Figures



**Figure A1.** Daily-averaged surface energy balance components, temperature and wind speed for R23p3 with respect to PROMICE and IMAU AWS observations on the GrIS for 2006-2015, for **(a)** shortwave downward (SWd) and **(b)** upward (SWu) radiation, **(c)** longwave downward (LWd) and **(d)** upward (LWu) radiation, **(e)** sensible heat flux (SHF), **(f)** latent heat flux (LHF), **(g)** 2-m temperature (T2m) and **(h)** 10-m wind speed (w10m). The 1-on-1 line is shown in black. Bias, root-mean-square error (RMSE), determination coefficient ($R^2$), intercept and slope of orthogonal total least squares regression, and number of points (N) are also shown.



*Author contributions.* CTvD led the model development, writing of the manuscript and analysis of model simulations. EvM and LHvU provided model support and SNG contributed to model development. MvT processed and provided the surface energy balance observations. WJvdB, MRvdB and TvdD contributed by analysing the results. All authors contributed to discussions on the manuscript.

*Data availability.* RACMO2.4 data for Greenland and Antarctica are available for the SMB components, SEB, near-surface temperature and
near-surface wind speed for 2006-2015. The data can be accessed here: https://doi.org/10.5281/zenodo.10854319 (Van Dalum et al., 2024). The IMAU AWS measurement are available from Smeets et al. (2022), and the PROMICE AWS measurements are available from How et al. (2022). The C3S Arctic snow depth observations (C3S, 2021) are available here: https://10.24381/cds.cf5f3bac. Data of the AntAWS are available from Wang et al. (2022), the QuikSCAT data from Trusel et al. (2013) and SMB observations from Machguth (2022).

*Competing interests.* At least one of the (co-)authors is a member of the editorial board of The Cryosphere. The authors have no other
competing interests to declare.

*Acknowledgements.* This work is supported by PROTECT. This project has received funding from the European Union's Horizon 2020 research and innovation program under grant agreement no. 869304, PROTECT contribution number TBD. We acknowledge the ECMWF for storage facilities and computational time on their supercomputer.



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
