# Peer review of "First results of the polar regional climate model RACMO2.4"

_EGUsphere, 2024_

## Author Comment (AC1)

Referee comment responses on the manuscript:

**First results of the polar regional climate model RACMO2.4**

by C.T. van Dalum et al.

We would like to thank the referees for their comments and we address them here. In black the comment, in orange the response, in blue the changes that we would implement in the manuscript. All line numbers refer to the old manuscript version.

**Review #1, from Oskar Landgren**

The authors present a thorough overview of the changes included in the updated version of the regional climate model RACMO, with many improvements tuned for the polar regions.

Evaluation includes comparison against comprehensive meteorological station datasets, but is otherwise mostly focusing on comparisons with the previous model version. While that indirectly (via articles referred to in the paper of evaluation of the previous version) enables comparison against remote sensing datasets, it would have been nice to see some figures where spatial patterns can be more closely examined for both model versions, for example against the mentioned CALIOP/CALIPSO dataset, or maybe MODIS or CLARA? Noting that the use of satellite data in the polar regions has its own issues (few near-pole overpasses, persistent cloud cover, lack of in-situ data for validation etc.) I think the current approach is fine.

You are right that several remote sensing products can still be used to evaluate the model. We are planning to apply more remote sensing products in upcoming publications, where we evaluate RACMO2.4 more thoroughly for the entire historical period for the Greenland and Antarctic ice sheet.

All in all, I find the manuscript well-written with a good structure and sound conclusions.

I have only a few minor comments before I can recommend accepting it:

L67-68: Please explain a little bit more what "flexibility" refers to in the sentence "recoded in order to improve code readability and flexibility". Does it for example include code refactoring, or make it easier to adapt to heterogeneous hardware in the future?

The coupling between HIRLAM and the IFS was recoded out of necessity, as the IFS code has changed considerably, and the previous coupling was not working anymore. As the IFS code is now 'object oriented', the code is now generally much more compact and more easy to read, as constants and variables are stored in several objects. The simplified code is therefore also more flexible to add new parameterizations and several input parameters, such as greenhouse gas concentrations, can now be read by simply providing a netcdf file. In addition, output writing is now much easier, improving user comfort. We have added to following to the manuscript to accommodate for this:

L67: In addition, it was necessary to recode the coupling between the HIRLAM dynamical core and the IFS physics module in order to accommodate to the new object-oriented code structure of the IFS, which simplifies the code by merging constants and variables into objects. Adding new parameterizations is therefore easier and code readability has improved. Furthermore, several parameters such as greenhouse gases can now be provided to the model more easily and writing of output is revised, improving user comfort by lowering the number of steps to add or change output.

L254-255: If I understood it correctly, you use fractional glacier cover. Does the sentence "Only measurement sites that are also located on a glaciated grid point in RACMO are included in this study" mean you are using cells with 100% glacier cover, or another threshold?

It is correct that we use fractional glacier cover now in RACMO2.4, but for the Greenland experiments presented in this study, we use the same glacier mask as is employed in RACMO2.3p3, which did not have fractional glacier cover. So in this case, a glaciated cell has 100% glacier cover. That we use the non-fractional ice mask of RACMO2.3p3 for the Greenland experiment is already mentioned on line 220-221

L265: Please motivate why the exclusion threshold is set as low as one missing hour per day. It sounds a bit aggressive to me, but perhaps there are very few days with only a few hours missing.

All days with at least one missing hourly measurement are not considered to remove the uncertainty when interpolating in between measurements, or when averaging partial data. This filter removes 460 days of data, or just 1.3% of the total 34 819 days of data when allowing for more than 5 hours of missing data per day or more (see attached Figure). We have included this information and hope that it becomes clearer in the revised manuscript.

[Figure]

Figure 1: "Total amount of valid daily AWS data from the selected AWSs between 2006-2015, for different thresholds of allowed missing data per day."

L265: This filter removes 460 days (1.3%) of the total dataset.

L342-343: MAR results are not shown for comparison. Consider being more explicit, for example "compare Fig. 3a with Fig. X of Fettweis et al. 2020", or adding "(not shown)", or dropping this sentence entirely.

We have changed the following to address this:

L342: … are qualitatively closer to those of MAR (see Fig. 4 of Fettweis et al. (2020)), which already…

I would personally prefer to have some of the figures use inverted colour scales, so that blue would indicate more water/snow/ice (in analogy with cooler temperature being illustrated with blue color) and red for less.

This applies to Figures 2, 3b, 8b and 10.

I can see that you may want to use red for positive changes for consistency (e.g. in Fig. 5 where you use the same colourbar for all panels), so I don't insist on you to take this comment into account.

We chose to show positive changes in red for the SMB and melt. To keep consistency, we decided to use this color configuration for all other difference plots. This is in particular important for a subfigure like Fig. 8b, where the other panels use the 'red is positive change' convention, and it would be confusing if it is not used consistently in this Figure and in the paper.

Typos:

L90: M in model should not be capitalised.

Done

L379: "such S6 or S7" -> "such as S6 or S7"

Done

L450: "Between -80 to -40 °C" -> "Between -80 and -40 °C"

Done

**Review #2**

In the paper by van Dalum et al., a thorough description is presented of the newest version of the polar regional climate model RACMO, version 2.4. The model is described, and a comparison is presented with the previous version and with observations.

The paper is very well written, the descriptions are thorough though still easily readable, and the results are well presented. It is recommended to be published after a very few adaptations.

Further, the results look very promising, particularly the effects of snow advection.

In Section 6, there are quantitative estimates of model bias for Arctic snow depth; it would improve the paper if some more qualitative considerations would be added: Is the current model good enough for intended uses related to this field? What are relative errors, etc. You might also consider to move this (Arctic) section before the Antarctic Section 5.

You are right that it is more logical to move the Section about the Arctic before the Antarctic. We have changed the following to accommodate for this:

L53: Additionally, a first impression of RACMO2.4 employed to the Arctic is shown in Sect. 5, where snow depth is discussed. Section 6 discusses the surface mass balance and temperature of Antarctica.

L457: We illustrate the capabilities of R24 to model the polar climate beyond the GrIS by analyzing modeled snow depth in a new pan-Arctic domain.

Moved L511-L514 to L506.

Qualitatively, the model output is very similar to findings reported in Arduini et al. (2019). Arduini et al. (2019) show major improvements in snow depth, snow water equivalent and the seasonal cycle of soil temperature by employing this multi-layer snow scheme. Temperature biases are also reduced compared to a single-layer model by taking a diurnal cycle more explicitly into account. Arduini et al. (2019) also concluded that the implementation of such a multi-layer snow scheme will likely improve the ECMWF ensemble forecasts by increasing the variability of near-surface weather parameters. polar regional climate models, however, have not been evaluated for seasonal snow depth in the Arctic region, making a comparison with such models difficult. We changed the following part of the text to accommodate for this:

L468: Arduini et al. (2019) present an analysis of the multi-layer snow scheme that is now also applied in R24. Qualitatively, the R24 snow depth is similar to findings of Arduini et al. (2019). They indicate that this scheme generally improves the snow depth, snow water equivalent and the seasonal cycle of soil temperature with respect to a single-layer snow scheme. Near-surface temperature biases are also reduced compared to a single-layer snow scheme by taking a diurnal cycle more explicitly into account. For central Eurasia, however, they show that snow depth is underestimated, possibly due to overestimated compaction or melt in forested areas or due to too little snowfall. For mountainous terrain in Norway and the Pacific coast of North America, they show a positive bias. It seems that these biases are inherited by R24.

L479: Nonetheless, Arduini et al. (2019) concluded that the implementation of such a multi-layer snow scheme in the IFS will likely improve the ECMWF ensemble forecasts by increasing the variability of near-surface weather parameters, and similar improvements are expected with R24. Other polar regional climate models similar to RACMO, however, have not been evaluated for seasonal snow depth in the Arctic region, making a comparison with such models difficult.

Minor comment: There seems to be no reference to Table 2 in the text even though numbers from the table are discussed.

We have added references to the table in several places. For example on line 398, line 403 and line 408